# Laser photogrammetry improves size and demographic estimates for whale sharks

Christoph A. Rohner[1,2], Anthony J. Richardson[2,3], Clare E.M. Prebble[1], Andrea D. Marshall[1,4], Michael B. Bennett[5], Scarla J. Weeks[6], Geremy Cliff[7,8], Sabine P. Wintner[7,8] and Simon J. Pierce[1,4]

[1] Marine Megafauna Foundation, Praia do Tofo Inhambane, Mozambique
[2] CSIRO Oceans and Atmosphere Flagship, Brisbane Queensland, Australia
[3] Centre for Applications in Natural Resource Mathematics (CARM), School of Mathematics and Physics, The University of Queensland, St Lucia Queensland, Australia
[4] Wild Me, Praia do Tofo Inhambane, Mozambique
[5] School of Biomedical Sciences, The University of Queensland, St Lucia Queensland, Australia
[6] Biophysical Oceanography Group, School of Geography, Planning and Environmental Management, The University of Queensland, St Lucia Queensland, Australia
[7] KwaZulu-Natal Sharks Board, Umhlanga, South Africa
[8] Biomedical Resource Unit, University of KwaZulu-Natal, Durban, South Africa

Corresponding author
Christoph A. Rohner, chris@marinemegafauna.org

## ABSTRACT

Whale sharks *Rhincodon typus* are globally threatened, but a lack of biological and demographic information hampers an accurate assessment of their vulnerability to further decline or capacity to recover. We used laser photogrammetry at two aggregation sites to obtain more accurate size estimates of free-swimming whale sharks compared to visual estimates, allowing improved estimates of biological parameters. Individual whale sharks ranged from 432–917 cm total length (TL) (mean ± SD = 673 ± 118.8 cm, $N = 122$) in southern Mozambique and from 420–990 cm TL (mean ± SD = 641 ± 133 cm, $N = 46$) in Tanzania. By combining measurements of stranded individuals with photogrammetry measurements of free-swimming sharks, we calculated length at 50% maturity for males in Mozambique at 916 cm TL. Repeat measurements of individual whale sharks measured over periods from 347–1,068 days yielded implausible growth rates, suggesting that the growth increment over this period was not large enough to be detected using laser photogrammetry, and that the method is best applied to estimating growth rates over longer (decadal) time periods. The sex ratio of both populations was biased towards males (74% in Mozambique, 89% in Tanzania), the majority of which were immature (98% in Mozambique, 94% in Tanzania). The population structure for these two aggregations was similar to most other documented whale shark aggregations around the world. Information on small (<400 cm) whale sharks, mature individuals, and females in this region is lacking, but necessary to inform conservation initiatives for this globally threatened species.

## INTRODUCTION

The whale shark *Rhincodon typus* (Smith 1828) is the world's largest fish species, measuring up to 2000 cm total length (TL) and 34 t in mass (*Chen, Liu & Joung, 1997*). Their large size,

tendency to spend much of their time at the surface (*Wilson et al., 2006*; *Brunnschweiler et al., 2009*; *Motta et al., 2010*) and predictable aggregative behaviour in certain coastal areas, make them susceptible to human threats such as directed fisheries (*Pravin, 2000*), boat strikes and net entanglement (*Speed et al., 2008*). Similar to most large sharks (*Cortés, 2002*), whale sharks are likely to grow and reach maturity slowly, leaving them vulnerable to depletion caused by human pressures (*Wintner, 2000*; *Cheung, Pitcher & Pauly, 2005*).

Whale sharks were listed as Vulnerable on the IUCN Red List of Threatened Species following rapid and substantial declines caused by targeted fisheries in the 1990s and early 2000s in the Indo-Pacific (*Norman, 2005*). Although a decrease in whale shark sightings may not necessarily indicate a decrease in actual whale shark numbers due to the highly mobile nature of these animals and variability in sighting conditions, studies that controlled for environmental factors in southern Mozambique (2005–2011; *Rohner et al., 2013*) and at Ningaloo Reef, Western Australia (1995–2004; *Bradshaw, Mollet & Meekan, 2007*) revealed substantial declines in sightings. This suggests that some aggregations in the Indian Ocean may have suffered population declines. Additional studies at Ningaloo Reef proposed that an apparent decline in mean length of whale sharks (*Bradshaw, Mollet & Meekan, 2007*) may have resulted from increased recruitment of smaller sharks to this location (*Holmberg, Norman & Arzoumanian, 2008*), rather than a decrease in survivorship of larger individuals (*Bradshaw, Mollet & Meekan, 2007*; *Bradshaw et al., 2008*). Such an interpretation would suggest that this regional population is recovering. These apparently conflicting results may be due partly to methodological differences among studies. *Holmberg, Norman & Arzoumanian (2008)*, *Holmberg, Norman & Arzoumanian (2009)* used mark-recapture population models and excluded transient sharks, whereas *Bradshaw, Mollet & Meekan (2007)* used demographic models, which are highly sensitive to variation in key biological parameters such as age or size at maturity. These parameters are poorly-known for whale sharks, and this high uncertainty decreases the predictive capability of demographic models (*Simpfendorfer, 1999*; *Bradshaw, Mollet & Meekan, 2007*). Determining life-history parameters is therefore crucial to improving whale shark management.

Generally, vertebral ageing studies are the source of most demographic data for elasmobranchs (*Cailliet et al., 2006*; *Pierce & Bennett, 2010*), but whale shark studies have been hampered by limited sample sizes and the difficulty in validating age results (*Wintner, 2000*). An alternative approach has been the use of growth rates on free-ranging sharks through the marking and recapture of individuals (*Pierce & Bennett, 2009*). In whale sharks, the common use of imprecise visual size estimation (*Rohner et al., 2011*) has precluded routine collection of growth data, and consequently long-term trends in mean lengths and growth should be interpreted cautiously.

Whale sharks show some degree of site fidelity (*Holmberg, Norman & Arzoumanian, 2009*; *Rowat et al., 2011*) that has allowed for basic biological parameters to be estimated through visual assessment, despite most aggregations being dominated by juvenile males. The length at which 50% of males reach maturity ($TL_{50}$) was estimated to be ~810 cm at Ningaloo Reef (*Norman & Stevens, 2007*), while growth rates were estimated to be

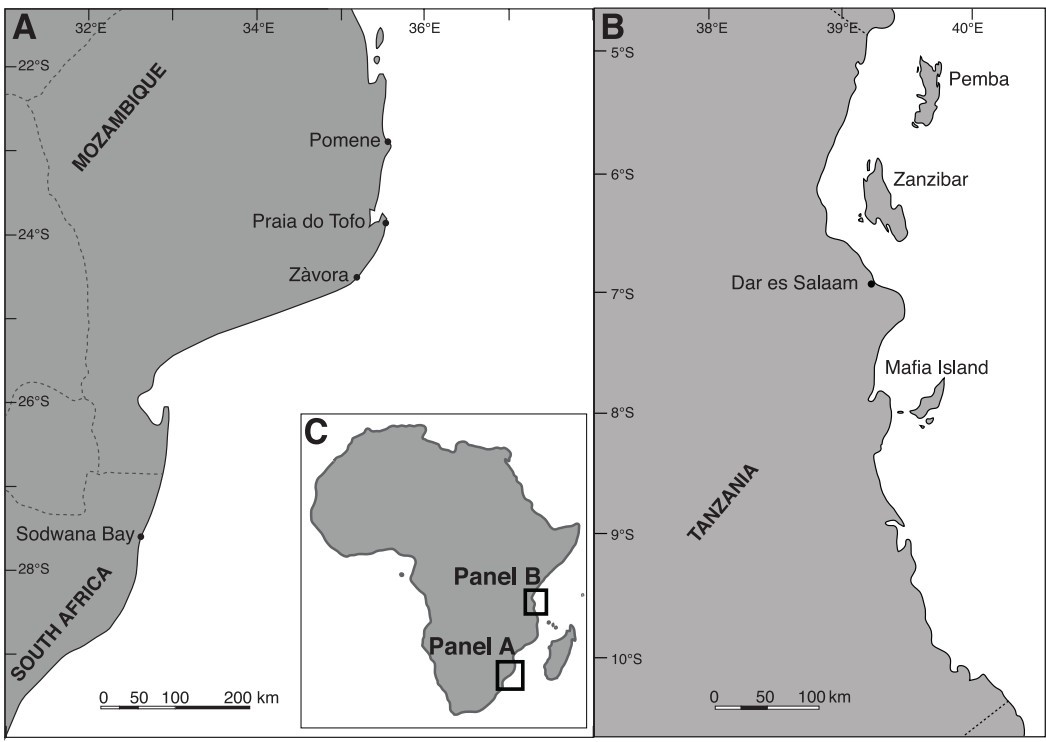

**Figure 1  Maps of the study locations.** The study locations off (A) Praia do Tofo in southern Mozambique and (B) Mafia Island in Tanzania, with (C) an inset of Africa for overview.

3–70 cm year$^{-1}$ in Belize (*Graham & Roberts, 2007*) and 45 cm year$^{-1}$ in the Maldives (*Riley et al., 2010*). However, visual size estimates can lack accuracy and precision, particularly where multiple observers are involved (*Holmberg, Norman & Arzoumanian, 2009*). By contrast, paired-laser photogrammetry (photogrammetry henceforth) is likely to be more accurate and precise (*Rohner et al., 2011*).

Here, we use photogrammetry to measure whale sharks at two coastal aggregation sites in the southwestern Indian Ocean; off Praia do Tofo (Tofo Beach) in southern Mozambique and off Kilindoni on Mafia Island, Tanzania. First, we aimed to describe the size ranges and sex ratios of sharks at these sites. Second, we aimed to assess length at which 50% of males reach maturity using total length and clasper length measurements from photogrammetry of free-swimming whale sharks in southern Mozambique and direct measurements of stranded individuals in northern South Africa. Third, we aimed to test whether photogrammetry can detect growth rates estimated over a 1–3 year time period.

## METHODS

### Study locations and whale shark searches

Photogrammetry data were collected from whale sharks off Praia do Tofo (23.85°S, 35.56°E) in southern Mozambique between January 2010 and October 2013 and off Mafia Island, Tanzania (7.90°S, 39.66°E) between October 2012 and December 2013 (Fig. 1). Whale sharks were spotted during boat-based searches (see *Pierce et al., 2010*),

and all data were collected while snorkeling alongside the sharks. Direct measurements of stranded sharks were obtained from Pomene, southern Mozambique (22.92°S, 35.56°E) and from the northern South African coast (~29.10°S, 31.64°E, Fig. 1). Unpublished photographic identification data (*Wild Me, 2014*) and satellite tagging results (*Rohner, 2013*) have demonstrated regular movements between northern South Africa and southern Mozambique, hence we treat them as a single population. Data collection in Mozambique was cleared by The University of Queensland's animal ethics committee (GPEM/184/12/MMF/SF) and research in Tanzania was approved by the Tanzania Commission for Science and Technology (COSTECH).

## Photographic identification

We first identified each whale shark that we measured with photogrammetry, by photographing the area behind the gills and above the pectoral fin (*Arzoumanian, Holmberg & Norman, 2005*). All photographs were taken with Canon G11/G12 compact digital cameras. The zoom function was not used. Identification photographs were submitted to the Wildbook for Whale Sharks library (www.whaleshark.org) and processed to assign a unique identity to each shark. Sightings were compared with images in the archived database of sharks to identify broader connectivity with other sites.

## Photogrammetry analysis

A laser photogrammetry system mounted on a housed digital camera, as described in *Rohner et al. (2011)* and in *Deakos (2010)*, was used to project two spots of green laser light (Sea Turtle Scuba Inc.; <5 mW power) onto the flank of each shark while a photograph was taken. Only sharks which we could measure photogrammetrically were included in all analyses. A photograph suitable for photogrammetric analyses needed to feature the flank region from fifth gill slit to the start of the first dorsal fin, both laser points had to be clearly visible, the shark had to be in a stretched position and the photograph had to be taken from the same horizontal level of the shark and with the laser pointers being at ~90° angle to the flank. Four observers took photogrammetry images, with the majority taken by CAR and SJP (~90%). Total length was extrapolated from a measurement of the flank between the 5th gill slit and the origin of the 1st dorsal fin ($B_{P1}$ in *Rohner et al., 2011*). This metric had a robust linear correlation for whale sharks between ~400–900 cm in length (*Rohner et al., 2011*). Where possible, multiple laser photogrammetric images were taken of the shark in each encounter to measure TL and improve the morphometric relationship between TL and the distance from the 5th gill slit to the origin of the 1st dorsal fin. All shark lengths are reported as total length unless otherwise specified.

## Assessment of the laser photogrammetry set-up

We followed the methods described in *Deakos (2010)* to assess the laser photogrammetry setup:

**Image distortion**: The airspace between the camera lens and the underwater housing refracts the incoming light and the shape of the lens itself can lead to image distortion. We thus quantified image distortion of the photogrammetry setup empirically underwater.

A grid of $10 \times 10$ squares was photographed. The pixels across the diagonal of the middle two squares were counted and this number was multiplied by 2, 3 and 4 to get the expected distance across 4, 6 and 8 squares, respectively. The observed and expected values (number of pixels) were plotted and a linear regression fitted to the data to obtain the image distortion function:

$$L_{\text{Expected}} = 1.0339 \times L_{\text{Observed}} - 31.516.$$

As the zoom on the camera was never used, this distortion function was constant (*Harvey & Shortis, 1998*) and was applied to all photogrammetry image measurements.

**Parallel alignment of lasers:** Lasers must be parallel to provide accurate data from varying distances to the target. The photogrammetry set-up was therefore regularly calibrated on land by measuring points 50 cm apart from 3, 5 and 8 m to the target. Photogrammetry images of whale sharks for size analysis were consistently taken at $\sim$4 m from the shark, so that the maximum tested distance (8 m) was about twice that used for size estimation, and errors would have been, on average, half as large.

**Parallax error**: Parallax error would lead to an underestimate of shark length if a photogrammetry image was not taken perpendicular to the target. The parallax error for our setup was assessed by measuring a 50 cm long object 5 times each from an angle of $10°$, $20°$, $30°$, $40°$ and $50°$. The percentage error was 2.9%, 8.3%, 16.6%, 27.5% and 39.1%, respectively. In the field, we had no means of estimating this angle for each photograph and thus correcting for potential parallax error. Instead, we exclude all images that appeared to be taken at $>10°$ angle.

Finally, the accuracy and precision of the photogrammetry setup were assessed by measuring a 258.6 cm pole 30 times underwater.

## Maturity assessment

The sex of each whale shark was determined visually by examining the pelvic fins for the presence of claspers, the external, paired reproductive structures of male sharks. Maturity in male sharks was assessed by examining the length and thickness of claspers (*Norman & Stevens, 2007*). Immature sharks have relatively small, thin claspers, and mature sharks have thick claspers that extend past the pelvic fins. Field observations suggest that whale shark claspers grow in length first before they get thicker. We classified sharks as mature only if their claspers were both long and thick, indicating calcification (Fig. 2). While clasper calcification in dead sharks is manually assessed, we think that calcified, stiff claspers of mature whale sharks can also be visually determined. Claspers of 46 male sharks from Mozambique and 22 sharks from Tanzania were measured using photogrammetry, while claspers from 11 males that were stranded along the northeastern coast of South Africa were measured directly. Clasper length (CL) was defined as the distance from the anterior end of the cloaca to the posterior tip of the clasper, equivalent to clasper inner length in *Compagno (2001)*. The TL and CL at which 50% of males were mature ($TL_{50}$ and $CL_{50}$) were each calculated using generalised linear models (GLM), with a binary logit function. We minimised potential differences among measurements

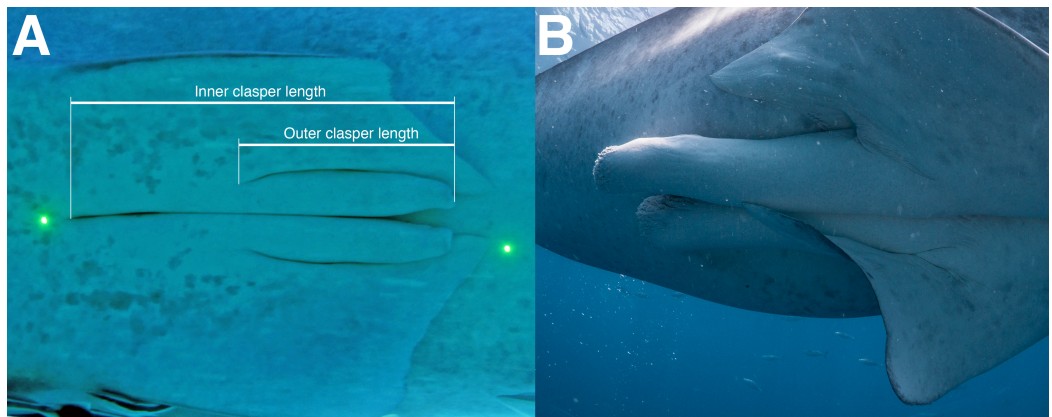

**Figure 2** Claspers of (A) an immature male; and (B) a mature male whale shark.

of live, free-swimming sharks and dead, stranded specimens by measuring natural TL (*Francis, 2006*) where possible, or scaling pre-caudal length (PCL) to TL based on a previously-derived morphometric relationship: $TL = 1.2182 * PCL + 33.036 (N = 41)$ in 4 of the 11 stranded sharks (*Wintner, 2000*; *Rohner et al., 2011*).

Three whale sharks were found stranded on 16 August 2009 at Pomene Beach in southern Mozambique (Fig. 1) and dissections were conducted on-site. The maturity status of the two female sharks was based on the condition of the ovary and the uteri, and of the male through examination of claspers, testes and accessory organs, similar to criteria in *Pierce, Pardo & Bennett (2009)*.

## Age determination

To determine whale shark age, we used growth patterns in the vertebral centra, as is done in most age and growth studies of elasmobranchs (*Cailliet et al., 2006*). Vertebrae anterior to the first dorsal fin were extracted from two of the stranded whale sharks from Mozambique; a 738 cm male and a 630 cm female. Vertebrae were stored frozen until x-radiography images were taken (Eklin EDR3 Mark III) to visualise band pairs following *Wintner (2000)* as a method of determining age. Some elasmobranchs lay down a pair of visually identifiable bands on their vertebrae as they grow (*Cailliet et al., 2006*). These band pairs consist of one opaque and one translucent band and represent summer and winter, respectively (*Cailliet et al., 2006*). We counted band pairs on two vertebrae from each shark. Three readers assessed each vertebra three times, independently of one another, after which the median was taken as a consensus count.

## Growth rates

We tested whether *in situ* photogrammetry had enough precision to determine growth rates of 13 whale sharks measured and subsequently re-measured over >340 days. Our growth rate estimates were compared to growth derived from back-calculated size at age data, assuming annual band pair formation. These data included band-pair counts from stranded sharks of known size from South Africa (*Wintner, 2000*) and from two of the sharks we dissected at Pomene, Mozambique. We produced a linear regression with 95%

**Table 1 Whale shark sizes in Mozambique and Tanzania.** The size of whale sharks measured with photogrammetry in Mozambique and Tanzania by sex, with male clasper measurements from South Africa included under Mozambique.

| | Total length (cm) | | | Clasper length (cm) | | |
|---|---|---|---|---|---|---|
| | N (%) | Mean (± SD) | Range | N | Mean (± SD) | Range |
| **MOZAMBIQUE** | | | | | | |
| Males | 87 (75.7%) | 692 (±119) | 445–934 | 57 | 54 (20) | 27–106 |
| Females | 28 (24.3%) | 670 (±108) | 439–858 | | | |
| Total | 123 | 684 (±118) | 439–934 | | | |
| **TANZANIA** | | | | | | |
| Males | 49 (87.5%) | 660 (±131) | 420–990 | 22 | 51 (15) | 31–89 |
| Females | 7 (12.5%) | 620 (±117) | 541–871 | | | |
| Total | 56 | 655 (±129) | 420–990 | | | |

confidence intervals (CI) from back-calculated size at age values. The zero value was set at 42 cm PCL following *Wintner (2000)*, as this is the approximate size of newly-born whale sharks (*Joung et al., 1996*; *Chang, Leu & Fang, 1997*).

# RESULTS

## Photogrammetry assessment
Length estimates of the 258.6 cm pole made with our photogrammetry equipment under controlled conditions were within a mean error of 1.2% or −3.2 cm. Lengths ranged from 254.7–256.6 cm and all measurements underestimated the true length. Precision was high, with a coefficient of variation of 0.17%.

## Morphometric relationship for TL
The morphometric relationship used for estimating TL in *Rohner et al. (2011)* was based on 27 data points, including 4 sharks from Japan. We updated this correlation here by including additional data from 14 fully-measured live sharks and by removing the 4 data points from outside our study region. The updated equation was:

$$TL = 4.902 BP1 + 72.579 (r^2 = 0.92, N = 37).$$

## Population structure
The 123 measured whale sharks in southern Mozambique ranged from 439–934 cm, with a mean ± SD of 684 ± 118 cm (Table 1). A significant sex bias was observed, with 75.7% male and 24.3% female in the 115 sharks for which sex was determined (Chi-square test, $\chi^2 = 26.420$, $P < 0.001$). Mean male size (range = 445–934 cm, mean ± SD = 692 ± 119 cm, $N = 87$) did not differ significantly from mean female size (range = 439–858 cm, mean ± SD = 670 ± 108 cm, $N = 28$) ($t$ test, $t = 0.67$, df = 49.65, $p = 0.506$), although all 6 sharks >860 cm were male (Fig. 3A).

The 56 whale sharks measured in Tanzania ranged from 420–990 cm, with a mean of 655 ± 129 cm (Table 1). A significant sex bias was present, with 87.5% male and 12.5% female

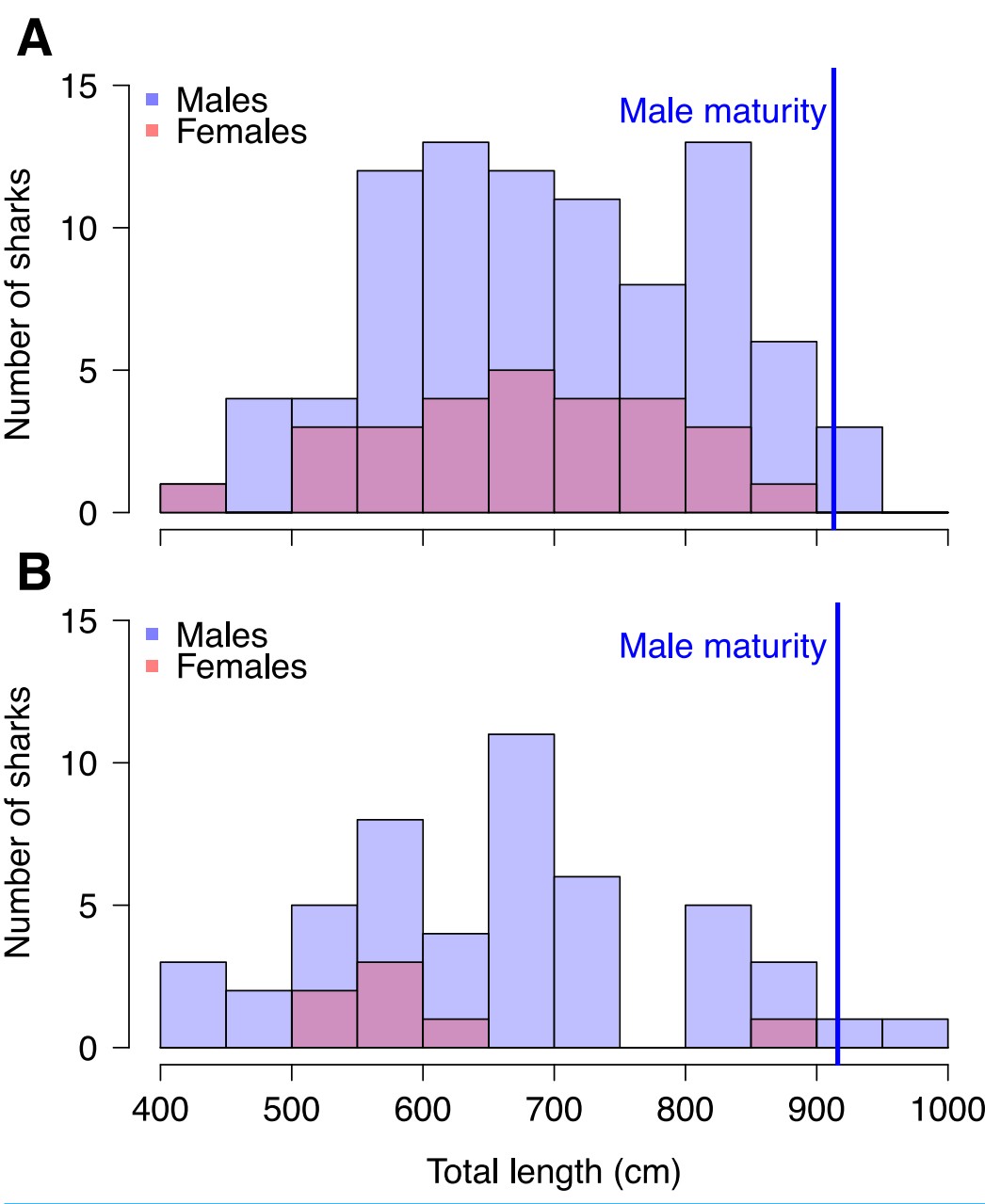

**Figure 3 Numbers of whale sharks per size bin in Mozambique and Tanzania.** The number of whale sharks (red = females, blue = males) per size bin in (A) Mozambique and (B) Tanzania.

in the 56 measured sharks for which sex was determined (Chi-square test, $\chi^2 = 56.3$, $P < 0.001$). The mean length of males ($660 \pm 131$ cm, $N = 49$) and females ($620 \pm 117$ cm, $N = 7$) were not significantly different ($t = 0.84$, df $= 8.32$, $p = 0.425$) (Fig. 3B).

## Size at maturity

Inner clasper lengths (CL) were measured for 46 sharks from Mozambique, 11 from South Africa and 22 from Tanzania. Eight sharks ranging from 823–1032 cm were mature and

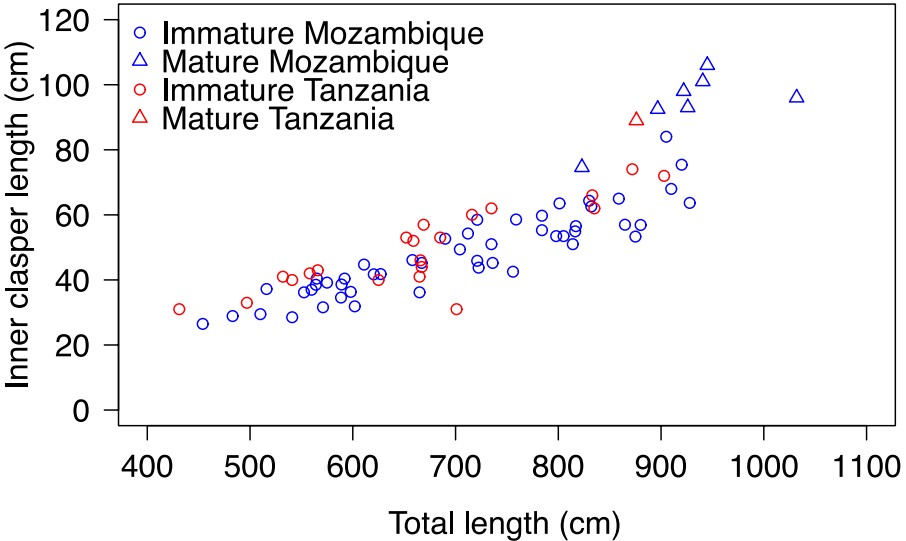

**Figure 4 Total length and clasper length for male whale sharks.** Total length and inner clasper length of male whale sharks ($\circ$, immature; $\triangle$, mature) in Mozambique and South Africa (blue) and Tanzania (red).

had clasper lengths ranging from 75–106 cm. The largest immature male was 928 cm and clasper length of immature males ranged from 26–84 cm (Fig. 4).

Based on the established connectivity between Mozambique and South Africa, we combined maturity data from stranded whale sharks in northeastern South Africa and Pomene (Mozambique) with data from whale sharks measured with photogrammetry in Mozambique. Maturity ($TL_{50}$) was attained at 916 cm (Residual Deviance = 19.9; p=0.012; AIC = 23.9), and $CL_{50}$ was 81.0 cm (Residual Deviance = 6.81; $p = 0.02$; AIC = 10.81; Fig. 5). One 876 cm mature male from Tanzania had a CL of 89 cm. It was slightly smaller than the largest immature shark (cf., 903 cm), but had longer claspers (cf., 74 cm). The 3 stranded sharks examined at Pomene measuring 738 cm (male), 630 cm (female) and 820 cm (female), were immature. The larger female had thin, strap-like uteri and a lattice-like ovary structure. No ovarian follicles were observed.

### Ageing and natural growth rates

Vertebrae of the 738 cm male and the 630 cm female had 26 and 22 band pairs, respectively. These data were added to the band pair counts of 15 whale sharks from Wintner (2000) to create an updated regression for band pair counts and length: $PCL = 22.44 * \text{band pairs} + 29.46 (r^2 = 0.99, N = 17)$.

Over the study period, we resighted 72% and 96% of measured individuals from Mozambique and Tanzania, respectively. Seven sharks from Mozambique and 24 sharks from Tanzania were measured multiple times over the study period, with a time gap of 3–1,068 days. Of these, 13 individuals were re-measured after more than 340 days had elapsed since the time of the initial size estimate and were used to examine growth. Mean growth rate was 5.6 cm year$^{-1}$ ($\pm 47.3$), with 6 sharks having decreased in length when re-measured (Fig. 6).

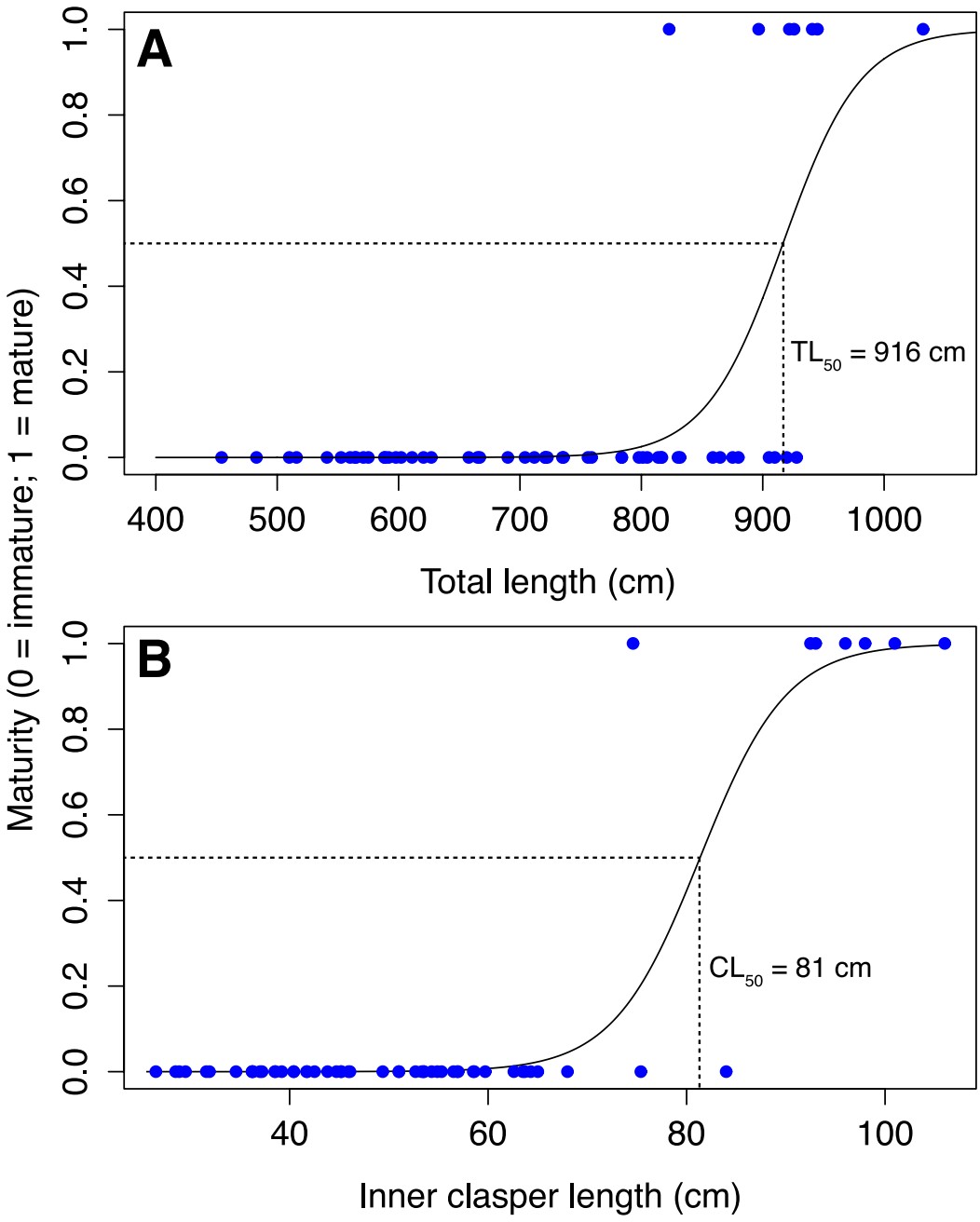

**Figure 5 Total length and clasper length at maturity.** Binary logistic plot of maturity in male whale sharks against (A) total length, with $TL_{50}$ = 916 cm; and (B) inner clasper length, with $CL_{50}$ = 81.0 cm.

## DISCUSSION

Photogrammetry improved the accuracy of whale shark size estimates. While the estimated error in visually-determined lengths of whale sharks was ∼10% (*Rohner et al., 2011*), our controlled tests of a stationary target showed that photogrammetry can greatly reduced this error. Precision was also high, with a CV of 0.17%, so length estimates were consistent across photographs. *Jeffreys et al. (2012)* also found high accuracy and precision

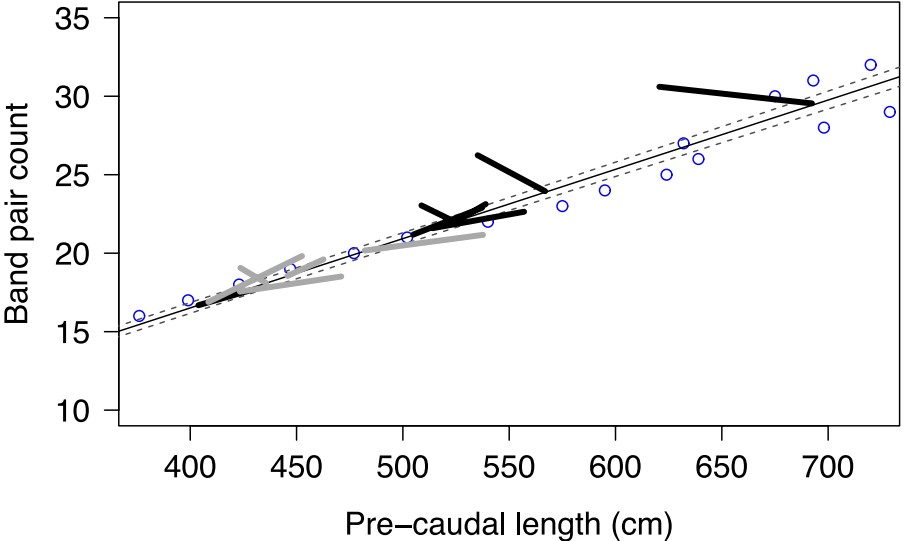

**Figure 6 Growth increments.** Observed growth increments of male (black) and female (grey) whale sharks plotted as size at age based on back-calculated lengths from vertebral band pair counts (*Wintner, 2000*) with 95% CI indicated by dashed lines. The initial size measurement was placed on the PCL/band pair count regression (dark line).

in experimental tests of a similar photogrammetry set-up. The major challenge with photogrammetry of whale sharks remains taking an image from the correct horizontal and vertical angle while the shark is in a straight, flexed position. Measuring only a portion of the body, such as PCL or $B_{P1}$, enhances precision as it excludes the caudal fin or the whole posterior part of the body which can be flexed when the shark is swimming and results in out of plane (foreshortened) images. We used $B_{P1}$ to scale TL in preference over the distance from the spiracle to the 5th gill slit (A1 in *Jeffreys et al., 2012*). This was because sharks in our study were mostly surface feeding, which resulted in a dorso-ventral flexion of the head that precluded an assessment of the A1 metric. Although the TL data used in our study are derived from a morphometric relationship between $B_{P1}$ and TL ($r^2 = 0.92$), our measurements are considered to be more accurate than those derived from visual estimates.

## Sex- and size-based segregation

Whale sharks measured in Mozambique and Tanzania exhibited pronounced sex- and size-based segregation. Most sharks were juvenile males of 550–850 cm, which is similar to other known whale shark aggregation sites in the Indian Ocean and elsewhere (Fig. 7 with references in the caption). Given that whale sharks can reach 2,000 cm (*Chen, Liu & Joung, 1997*), the size structure observed in these aggregations show that only a proportion of a whale shark population is seen at these coastal sites. Mean sizes of 684 cm in Mozambique and 640 cm in Tanzania were considerably larger than that recorded from Djibouti, Saudi Arabia, Taiwan and inshore sites in the Gulf of California. These sites appear to be dominated by small juveniles, while a larger size range of juveniles is present in Mozambique and Tanzania. By contrast, whale sharks here were considerably

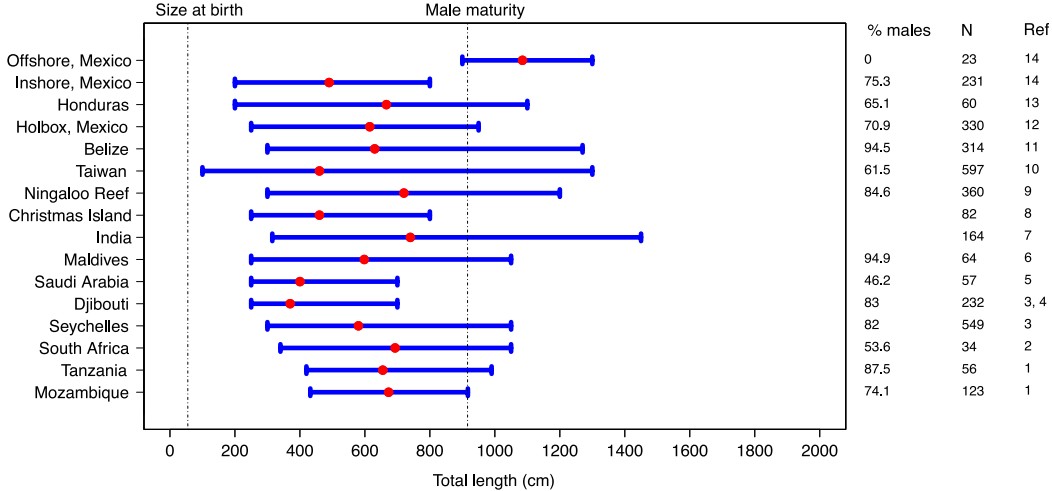

**Figure 7 Comparisons of whale shark aggregation sites.** The population structure of whale shark aggregations around the world, with mean total length in red and length range in blue, plotted on the total length range for the species. References: 1 (this study), 2 *Beckley et al., 1997*, 3 *Rowat et al., 2011*, 4 *Brooks et al., 2010*, 5 *Berumen et al., 2014*, 6 *Riley et al., 2010*, 7 *Pravin, 2000*, 8 *Hobbs et al., 2009*, 9 *Norman & Stevens, 2007*, 10 *Hsu, Joung & Liu, 2012*, 11 *Graham & Roberts, 2007*, 12 *Ramírez-Macías et al., 2012*, 13 *Fox et al., 2013*, 14 *Ramírez-Macías, Vázquez-Haikin & Vázquez-Juárez, 2012*.

smaller than at offshore sites in the Gulf of California where mostly large, mature females are seen (Fig. 7 with references in the caption). Size ranges of 439–934 cm observed in Mozambique and 415–971 cm in Tanzania were smaller than reported for most other locations, although this may partly be a consequence of the improved precision of size estimates from photogrammetry in comparison to visual estimates and the comparatively short time-frame of this study.

A male sex bias is common at monitored whale shark aggregation sites. The percentage of male sharks in Mozambique (76%) was similar to northeastern South Africa (73%) and inshore sites in the Gulf of California (75%), but lower than that in Tanzania (88%), the Maldives (95%), Djibouti (83%), Ningaloo Reef (85%) or the Seychelles (82%) (Fig. 7). By contrast, the coastal aggregation in Saudi Arabia had about equal numbers of juvenile males and females, whereas offshore sites in the Gulf of California and the Galapagos Islands mainly had large females (*Ketchum, Galván-Magaña & Klimley, 2012*; *Ramírez-Macías, Vázquez-Haikin & Vázquez-Juárez, 2012*; *Hearn et al., 2013*; *Berumen et al., 2014*). The apparent sex bias and the narrow size range of whale sharks across the Indian Ocean aggregation sites raises intriguing questions concerning the location of newborn, female, and larger mature sharks. Whale sharks are born at ∼45–60 cm (*Joung et al., 1996*; *Chang, Leu & Fang, 1997*), but <250 cm individuals are rarely seen anywhere in the world and there are only 19 reports of sharks <150 cm (*Rowat & Brooks, 2012*). The sex ratio of whale shark embryos was almost equal (1: 0.98 females to males, $N = 297$) in the only pregnant whale shark investigated to date (*Chang, Leu & Fang, 1997*). *Chang, Leu & Fang (1997)* found no inter-sex difference in the length or mass of embryos and hence female neonates are assumed to have similar survival rates to males. The pronounced

segregation in most coastal whale shark aggregations suggests that whale sharks occupy different habitats, or use the same habitats differently, depending on their sex and size.

While the sex bias and the predominance of immature whale sharks at coastal sites could conceivably be an artifact of the previous targeted fisheries activities in the Indian Ocean and Western Pacific, there are several arguments against this being the case in Mozambique and Tanzania. First, there appears to be little or no connectivity among the largest whale shark aggregations in the Indian Ocean (*Wilson et al., 2006*; *Brooks et al., 2010*; *Sleeman et al., 2010*). This suggests that fisheries in the Maldives, India, or further away in Taiwan and the Philippines should not have affected the population structure in the Western Indian Ocean, although they may have led to declines in the east at Ningaloo Reef (*Bradshaw, Mollet & Meekan, 2007*; *Bradshaw et al., 2008*) and off Thailand (*Theberge & Dearden, 2006*). Second, evidence suggests that the majority of sharks caught in fisheries were males or juveniles. Most sharks landed in Taiwan were juvenile males (*Hsu, Joung & Liu, 2012*). A large proportion of the catch from India contained immature sharks, though the sex of the sharks was not reported (*Pravin, 2000*). Interviews with fishers and catch records from the Philippines also indicated that landed sharks were largely immature, again with no information on the sex ratio (*Alava & Dolumbalo, 2002*). Last, coastal whale shark aggregations in and around the Caribbean Sea, where there is no history of fishing for whale sharks, are also dominated by immature male sharks (*Graham & Roberts, 2007*; *Ramírez-Macías et al., 2012*; *Fox et al., 2013*). Therefore, data suggest that juvenile and male dominated whale shark aggregations are natural and not necessary an artifact of selective fishing pressures.

Segregation is common in many shark species, with populations usually divided socially and/or geographically into units of sub-adults, mature males and mature females (*Springer, 1967*; *Klimley, 1987*; *Richardson et al., 2000*; *Bansemer & Bennett, 2011*). This is thought to be due to differences in diet or swimming capabilities or to reduce intra-specific competition, aggression and predation (*Springer, 1967*; *Wearmouth & Sims, 2008*). The reason for the prevalence of juvenile male whale sharks at known aggregation sites is unclear, and although different diet preferences for juveniles and adults has been suggested (*Ketchum, Galván-Magaña & Klimley, 2012*), this does not explain the sex bias. The segregation observed in Mozambique, Tanzania and elsewhere indicates that larger individuals and neonates use different habitats than juveniles. Similarly, mature sharks of both sexes are not often seen at coastal sites and may be completely oceanic. Although few data are available from the Indian Ocean, large mature sharks are regularly seen in the open ocean in other areas (*Hazin et al., 2008*; *Hearn et al., 2013*; *Afonso, McGinty & Machete, 2014*). Their larger size and superior swimming efficiency may enable them to move further horizontally and vertically and thus forage more successfully in a patchy offshore prey landscape (*Sims et al., 2006*).

## Size at maturity

Our $TL_{50}$ of male whale sharks was 916 cm, $\sim$100 cm larger (13% of total length) than that visually estimated for Ningaloo Reef sharks (*Norman & Stevens, 2007*), and $\sim$200 cm

larger (24% of total length) than those off the Yucatan coast of Mexico (*Ramírez-Macías et al., 2012*). These large differences are potentially significant, and suggest genuine biological differences among sharks using these sites. Regional differences among sizes at maturity or life-history traits of elasmobranch species are not uncommon, and have been documented in bonnethead sharks *Sphyrna tiburo* (*Lombardi-Carlson et al., 2003*), greeneye spurdog shark *Squalus mitsukurri* and porbeagle sharks *Lamna nasus* (*Francis & Duffy, 2005*), and cownose rays *Rhinoptera bonasus* (*Neer & Thompson, 2005*), among others. Evidence from mitochondrial and microsatellite DNA showed that Atlantic and Indo-Pacific whale sharks never or rarely mix, while no evidence of stock structure was found within the Indian Ocean (*Vignaud et al., 2014*). The marked difference in $TL_{50}$ between Mozambique and the Yucatan coast of Mexico thus is consistent with these genetic results. It is unclear whether our photogrammetric results are directly comparable with the visual size estimates from Ningaloo Reef (*Norman & Stevens, 2007*) due to the differing methods employed. While there are no genetically distinct stocks of whale sharks within the Indian Ocean (*Vignaud et al., 2014*), photo-matching (*Brooks et al., 2010*) and tracking studies (*Wilson et al., 2006*; *Sleeman et al., 2010*) have not demonstrated any interchange between the eastern and western Indian Ocean populations. There may thus be population differentiation among Indian Ocean whale shark aggregations on a shorter time-scale than detected in genetic studies. A significant size-at-maturity difference between the eastern and western Indian Ocean, if it does exist, would suggest population-level separation within this ocean basin.

Mature female whale sharks are rarely observed and all sharks >900 cm observed in our study were male. While it is impossible to assess maturity in females externally, in the absence of visible pregnancy, females of 820 cm (this study), 870 cm (*Beckley et al., 1997*) and 880 cm (*Pai, Nandakumar & Telang, 1983*) examined in the Indian Ocean were immature. The only directly-measured mature female to date was 1060 cm (*Joung et al., 1996*), and mature females in the Gulf of California were visually-estimated at 900–1300 cm (*Ramírez-Macías, Vázquez-Haikin & Vázquez-Juárez, 2012*). Potential stock differences notwithstanding, this suggests that none of the females in our study were mature.

## Applicability of laser photogrammetry to estimating growth rate

Laser photogrammetry is clearly an improvement on the standard method of estimating length visually, but our results indicate that it is not sufficiently precise for measuring growth rates of individuals over short time spans (a few years) relative to their ∼80 year lifespan (*Hsu et al., 2014*). The negative growth rates we estimated are likely to reflect this limitation. Although we demonstrated the accuracy of measurement for a static target of known length, it was not possible to document the accuracy of the technique when applied to a free-swimming whale shark, as there was no way of knowing the true length of the subject shark. It would be best to make repeat measurements on the same individuals on the same day to evaluate precision, although this was not possible in our study because we could not re-measure individuals on the same day. Our summary is that laser photogrammetry is useful for routine length measurement of whale sharks, but

should not be used for estimating growth rates over short (1–3 year) periods. However, it may remain useful for measuring growth over long periods (decades).

In conclusion, laser photogrammetry estimates are likely to be more accurate and precise than visual estimates of length and size at maturity, but we suggest that they are not used for growth rate estimates over short time periods. Accurate measurement of life-history parameters can improve demographic models for the whale shark and thus facilitate better assessment of its vulnerability to fishing pressures or recovery from population declines. We also show that the size range and sex ratio of whale sharks from Mozambique and Tanzania are similar to those at most other aggregation sites globally, in that the population consisted largely of ~450–950 cm juvenile sharks, most of which were males. The observed population segregation by size and sex reinforces the need to determine the whereabouts of young-of-the-year and small juvenile sharks, immature female sharks, and mature sharks of both sexes to improve conservation and management for this globally threatened species.

## ACKNOWLEDGEMENTS

We thank P Bassett, Marine Megafauna Foundation volunteers and staff and All Out Africa volunteers for assistance with fieldwork in Mozambique. Casa Barry Lodge and Peri-Peri Divers provided field support in Mozambique. We thank Jason Rubens, Haji Machano, Jesse Cochran and Fernando Cagua for facilitating fieldwork in Tanzania. Liberatus Mokoki and Jean and Anne de Viliers provided field support in Tanzania. We also thank N Ayliffe for help with whale shark dissections in Pomene and the Baker & McVeigh Equine Hospital in Summerveld, KZN for taking x-radiography images of the vertebrae. We thank the Editors David Johnston and John Bruno as well as Mark Deakos and two anonymous reviewers for their comments on the draft manuscript, which have considerably improved our paper.

### Funding

This study was supported by the Shark Foundation, GLC Charitable Trust, Rufford Small Grants (grant 23.12.08), Project AWARE International, Ocean Revolution, Fondation Ensemble, WWF Tanzania (grant CN74) and one anonymous donor. The funders had no role in study design, data collection and analysis, decision to publish, or preparation of the manuscript.

### Grant Disclosures

The following grant information was disclosed by the authors:
Shark Foundation, GLC Charitable Trust, Rufford Small Grants: 23.12.08.
Project AWARE International, Ocean Revolution, Fondation Ensemble, WWF Tanzania: CN74.

**Peer**J

## Competing Interests

Christoph A. Rohner, Clare E.M. Prebble, Andrea D. Marshall and Simon J. Pierce are employees of Marine Megafauna Foundation. Anthony J. Richardson is an employee of CSIRO. Simon J. Pierce and Andrea D. Marshall are employees of Wild Me.

## Author Contributions

- Christoph A. Rohner and Simon J. Pierce conceived and designed the experiments, performed the experiments, analyzed the data, contributed reagents/materials/analysis tools, wrote the paper, prepared figures and/or tables, reviewed drafts of the paper.
- Anthony J. Richardson analyzed the data, contributed reagents/materials/analysis tools, wrote the paper, prepared figures and/or tables, reviewed drafts of the paper.
- Clare E.M. Prebble performed the experiments, contributed reagents/materials/analysis tools, wrote the paper, reviewed drafts of the paper.
- Andrea D. Marshall conceived and designed the experiments, contributed reagents/materials/analysis tools, wrote the paper, reviewed drafts of the paper.
- Michael B. Bennett and Scarla J. Weeks contributed reagents/materials/analysis tools, wrote the paper, reviewed drafts of the paper.
- Geremy Cliff and Sabine P. Wintner performed the experiments, contributed reagents/materials/analysis tools, wrote the paper, reviewed drafts of the paper.

## Animal Ethics

The following information was supplied relating to ethical approvals (i.e., approving body and any reference numbers):

Research was cleared by The University of Queensland's Animal Ethics clearance number GPEM/184/MMF/SF.

## Field Study Permissions

The following information was supplied relating to field study approvals (i.e., approving body and any reference numbers):

Research in Tanzania was approved by the Tanzania Commission for Science and Technology (COSTECH).

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
