# Peer review of "Laser photogrammetry improves size and demographic estimates for whale sharks"

_PeerJ, doi:10.7717/peerj.886_

## Round 0.1 · original submission · Major Revisions

After receiving three reviews of the paper I am returning it to you for major revisions. All three reviews were almost entirely positive about the manuscript, but two reviewers raised issues about how the analysis addressed error, both for the measurement technique and for observer estimates. In particular, both Reviewer 2 and 3 indicated that and assessment of multiple photos would be valuable to better assess growth rates, and at longer time scales. These reviewers also suggested that more detail on the use of clasper morphology would be useful for the PeerJ audience, and they also indicated that the conclusions reached in the MS may overstretch aspects of the analysis and results. In particular, Reviewer 3 provided a detailed edit of the manuscript that provides excellent editorial advice as well as suggestions on how to deal with refined methods, more rigorous analysis and to better cite previous works in this area. Reviewer 2 also indicated that a small component of a revised manuscript could also further build on studies employing similar techniques for different taxa.

Reviewer 1 ·

Basic reporting

No comments.

Experimental design

No comments.

Validity of the findings

No comments.

Additional comments

Well rounded paper. Congratulations on adding significance and precision to a technique that is widely used in the field.
The science is coherent and properly analysed. As the authors carefully explored, basic life history of the whale shark is lacking, and the analysis of the size at maturity adds a great deal to the literature, and its implications for conservation. On that note, line 171-172, maybe elaborate in the discussion on clasper morphology of mature individuals as observed in this study?
Even though previous work (Rohner et al., 2011; Jeffreys et al., 2012) put some focus on the relationship of using different parts of the body to scale the total length of the animals, I wondered if this was explored as part of the accuracy of the technique given that it was a big sample size?
Great paper and carefully structured.

Reviewer 2 ·

Basic reporting

The paper is generally clearly written, the work is presented in the context of previous studies, and confirms to journal style.

As a non-shark biologist - some additional background to the age determination would be useful to understand the methodology based on a analysis of band pairs.

Experimental design

Aspects of the methodology require further detail to be reproducible.

l 118 - info on manufacturer and power of lasers required.
l 127 - " suitable photogrammetry image" - how is this determined?
l 138-144 - I find this part of the methodology very difficult to follow - does it really show whether or not distortion is equal across the whole field of view?
l 162 - how does one evaluate whether images were at a >10 degree angle?
l168 - Clasper length seems to have been used to assess maturity - yet the size ranges of mature and immature claspers overlaps (l240)- how does this work?

And the analysis should be more vigorous

l 131 - multiple pictures were taken but there appears to have been no attempt to conduct repeat estimates and estimate measurement error.
l262 the use of repeat measurements made just 3 days apart to assess growth rates seems unwise - especially given that measurement errors from repeat photographs taken within a day are not reported. I would suggest restricting this analysis to picture taken eg 1 year apart??

Validity of the findings

l 211 suggests that the results were accurate, yet the mean error suggests there was a bias.

l212 - given the range of lengths, the CV seems high - but all this is difficult to assess without more information on measurement errors derived from repeat photographs.

l362 - can we really conclude too much about these population differences in length given the differences in technique - especially as some of the published studies were simply visual estimates. Instead, why not highlight as a hypothesis that could be tested using these more quantitative techniques.

Additional comments

It would be worthwhile bringing in discussion about how laser photogrammetry has been used in other species, and perhaps compare approaches, accuracy and errors?

·

Basic reporting

Much of the sentence structure needs some work, I made suggestions in the attached document. I added comments in areas where additional explanation was needed to provide more context to the reader.

Experimental design

The aims of the study are clearly defined in the introduction. The connection between obtaining accurate measurements of sharks, population sex ratios and growth rates pertain to management of these shark populations could be strengthened. Some of the methods section mimic very closely a paper that was not cited, that citation was suggested. Comparisons are made between the percent error of visual estimation methods and laser-photogrammetry methods demonstrating one is better than the other. But in order to get accuracy, you need to know the actual size of the target, which can only occur with a fixed object of known size. Since the visual estimates were done on free-ranging animals, not sure how a percent error can be obtained unless it is across observer error, which is very different than accuracy. Having divers visually estimate the size of a pipe in the water and compare those estimates to the laser-photogrammetry methods would seem like a more accurate way to compare the two techniques. More comments relating to this in the attachment.

Validity of the findings

The paper is strong in the sense of having relatively large samples sizes, having dead animals to ground truth morphometric proportions and age based on TL, in addition to other literature with estimated metrics, which helps with the robustness of the conclusions. However, I found some conclusions to be over-reaching and comments related to this are in the attachment. Seems the discussion section could be improved by minimizing the reporting of results and provide assessment as to why the results did or did not match expectations or findings from other research. Suggestions can be found in the attachment.

Additional comments

I applaud the author for the amount of field work that has gone into this manuscript. I believe that after restructuring some of the text and addressing some the comments, this paper has a lot of valuable content that can be beneficial to others working in this area of research. I look forward to the final product.

---

## Round 0.2 · Major Revisions

Although two of reviewers were happy with the revisions you have made and feel the manuscript is essentially ready for publication, reviewer #2 still has a significant concern about the repeatability within a single time period. Can you please respond to this concern, and either make the change the reviewer asks for or explain why you think it is not necessary.

Reviewer 1 ·

Basic reporting

No comments.

Experimental design

No comments.

Validity of the findings

No comments.

Additional comments

The authors have greatly improved the manuscript by thoroughly addressing the reviewers' comments. This manuscript has applications to strengthen whale shark research at different sites by improving demographic data. The methods can also be adapted for work with different species. We congratulate the authors for this valuable contribution to the field.

Reviewer 2 ·

Basic reporting

MS now much clearer

Experimental design

I still feel the information presented on individual rates of growth (and shrinking) is meaningless unless information is given on repeatability within a single time period. I suggest that this information is dropped until the repeatability study has been done, and the data are restricted to the first measurement made for each individual - and the measurement of a pole can provide give an indication of the approximate validity of single measurements.

Validity of the findings

Good - with the above caveat

·

Basic reporting

No comments.

Experimental design

No comments.

Validity of the findings

No comments.

Additional comments

I commend the authors on thorough responses to the reviewer comments. Other than two small items listed below, I look forward to the publication of this manuscript.

Lines76-77: “have suffered population declines” should be changed to “may have suffered population declines” since animals no longer visiting those regions could also explain a decline in sightings.

Line 201: Deakos (2010) needs to be added to the references.

Thank you for the opportunity to review this manuscript.

Mark Deakos

---

## Round 0.3 · accepted · Accept

Thank you for your explanation, re, Reviewer 2's concerns about repeatability. I believe your edits to the Discussion make your recommendations clear and acknowledge the potential limitations highlighted by Reviewer 2.

I believe the ms is ready for publication.